

# *Ampelisca eschrichtii* Krøyer, 1842 (*Ampeliscidae*) of the Sakhalin Shelf in the Okhotsk Sea starve in summer and feast in winter

Valentina B. Durkina[1,*], John W. Chapman[2,*] and Natalia L. Demchenko[3,*]

[1] Laboratory of Physiology, National Scientific Center of Marine Biology FEB RAS, Vladivostok, Russia
[2] Department of Fisheries and Wildlife, Oregon State University, Newport, OR, United States of America
[3] Laboratory of Marine Ecosystem Dynamics, National Scientific Center of Marine Biology FEB RAS, Vladivostok, Russia
[*] These authors contributed equally to this work.

## ABSTRACT

**Background**. *Ampelisca eschrichtii* Krøyer, 1842 of the Sakhalin Shelf of the Okhotsk Sea, Far Eastern Russia, comprise the highest known biomass concentration of any amphipod population in the world and are a critically important prey source for western gray whales. Growth and reproduction in this population has not been apparent in summer. However, they are not accessible for sampling in winter to test a previous default conclusion that they grow and reproduce in winter.

**Methods**. We tested the default winter growth and reproduction hypothesis by detailed comparisons of the brood and gonad development among 40 females and 14 males and brood sizes among females observed since 2002. Our test included six predictions of reproductive synchrony that would be apparent from gonad and brood morphology if active reproduction occurs in summer.

**Results**. We found high prevalences of undersized and damaged oocytes, undersized broods, a lack of females brooding fully formed juveniles, atrophied ovaries, and males with mature sperm but lacking fully developed secondary sex morphologies required for pelagic mating. All of these conditions are consistent with trophic stress and starvation.

**Discussion**. These *A. eschrichtii* populations therefore appear to starve in summer and to grow and reproduce in winter. The Offshore *A. eschrichtii* populations occur in summer below water strata bearing high phytoplankton biomasses. These populations are more likely to feed successfully in winter when storms mix phytoplankton to their depths.

# INTRODUCTION

The densest and highest biomass populations of gammaridean amphipods known in the world consist primarily of *Ampelisca eschrichtii* Krøyer, 1842 and occur at 40–60 m depths in the Offshore feeding grounds of the critically endangered western gray whale, *Eschrichtius robustus* (Lilljeborg, 1861) (*IUCN, 2008*) on the northeastern Sakhalin Island

Corresponding author
Valentina B. Durkina,
vdurkina@mail.ru

Shelf at approximately 52.0°N and 143.7°E (*Demchenko et al., 2016*). The production and growth of these populations are of international concern for gray whale conservation and for understanding high latitude benthic ecosystem dynamics. Estimates of the productivity of these populations have remained complicated due to irregular and seldom replicated sampling over time within years and due to the absence of any sampling between late fall and early spring (winter from here on). *Demchenko et al. (2016)* partially solved this problem by integrating comparisons of *A. eschrichtii* length density modes and female brood development stages between late spring and early fall (summer from here on) among six sampling years between 2002 and 2013. They discovered that Sakhalin Shelf *A. eschrichtii* are gonochoristic, iteroparous, mature at body lengths greater than 16 mm, have a predominantly two-year life span and a low incidence of individuals that survive to three years.

*Demchenko et al. (2016)* noted also that brooding females were rare in their summer samples and that females brooding undifferentiated embryos or bearing juveniles ready for release were absent. *Demchenko et al. (2016)* noted in addition that terminal phase reproductive males were absent and that length density modes in their populations did not increase over time. *Demchenko et al. (2016)* included preliminary histological analyses that additionally revealed vitellogenic oocytes appearing to be undergoing lysis and resorption (atresia), a common symptom of spent or starving fish, decapods and amphipods (*Sheader, 1996*; *Sainte-Marie, 1991*; *Kurita, Meier & Kjesbu, 2003*; *Santos et al., 2005*; *Santos et al., 2009*). Lysed oocytes are thus signs of food limitation, starvation, reproductive failure and, by default, evidence that the Offshore *A. eschrichtii* do not grow or reproduce in summer. *Demchenko et al. (2016)* therefore concluded that *A. eschrichtii* reproduction must occur in winter.

*Demchenko et al.*'s (*2016*) samples covered sufficient spans of years and months over summer to preclude one time occurrences of starvation effects. Extended survival of individuals that cannot later reproduce would be an evolutionary conundrum. In contrast, atresia of reproductive cells in poor trophic conditions is likely to be adaptive if the result is greater survival and reproduction later.

Possible contradictions to *Demchenko et al.*'s (*2016*) winter production hypothesis, that might have indicated summer reproduction, included 3.8 mm length (0-age) juveniles in their samples and females bearing broods. Moreover, their histological sample, consisting of 8 reproductive size females collected in October 2013, was limited numerically and temporally. *Demchenko et al.*'s (*2016*) hypothesis of winter growth and production is also in contrast to previous reports of summer growth and production in North Pacific ampeliscid populations (*Coyle et al., 2007*) and to previous conceptions of high latitude benthic production occurring mainly in summer. *Demchenko et al.*'s (*2016*) new hypothesis thus warrants close examination. The Offshore area in winter however, is covered by pack ice and frequented by severe storms that prevent ship access needed for benthic sampling (*Fadeev, 2012*). Direct winter sampling of the Offshore area that would permit direct tests for winter growth and reproduction has therefore not been possible. Additional tests of *Demchenko et al.*'s (*2016*) default winter production hypothesis are therefore restricted to increasingly detailed examinations of growth and reproduction in summer that we address here.

High survival on trophic reserves during periodically low food resources is consistent with amphipod and crustacean life histories (*Lawrence, 1976*). Lipids are a major energy reserve in aquatic invertebrates (*Parrish, 2013*) including amphipods (*Sainte-Marie, 1991*) and also major components of egg production (*Charniaux-Cotton, 1985*). We compared reproductive morphologies of *A. eschrichtii* and the sizes, conditions, maturity and synchrony of their reproductive cells in summer to more clearly resolve whether food limitation and reproductive failure are likely to occur in the Offshore populations in summer.

The synchrony of oocyte growth and development in amphipods begins with the transformation of females from *F0* to *FI* when they extrude oocytes from their ovaries into the marsupium. Except in conditions of extremely high and uneven mortality, which have not been reported, the frequencies of sequential reproductive stages within populations over extended periods must coincide with their durations. We thus expect rapid replacement of *FIII* juveniles with fresh broods of embryos and few *FIV* females in actively reproducing populations. The absence of *FIII* broods in actively reproducing populations is therefore not expected when *FIV* females are present.

Terminal phase reproductive males of all *Ampelisca* species develop enlarged antennae and pleosomites and a dorsal keel on the urosome. These morphologies are adaptations that must occur in synchrony with pelagic mating (*Hastings, 1981*; *Borowsky & Aitken, 1991*).

The ontology and maturation of spermatophores in *Ampelisca* males and the formation and development of female *Ampelisca* oocytes are apparent from histology (*Hastings, 1981*; *Johnson, Stevens & Watling, 2001*). Amphipod females produce oogonia from mitotic division of primary oogonia which develop into oocytes through meiosis. The vitelogenic oocytes grow into lipid rich oocytes (*Charniaux-Cotton, 1985*). Females extrude the mature oocytes through two ventral oviducts of pereonite 5 into an external marsupium immediately after molting while the new exoskeleton remains flexible (*Hyne, 2011*). Amphipod oocytes become embryos after they are fertilized from spermatophores, which males deposit in the marsupium at the same time as arriving oocytes (*Johnson, Stevens & Watling, 2001*). The biomass and energy reserves of mature oocytes therefore must equal or exceed the biomass and energy reserves of viable embryos. Oocytes lacking the critical embryo biomass are, in turn, incompetent for reproduction. The lecithotrophic amphipod embryos develop without additional nourishment from the parent and then hatch and emerge from the marsupium fully formed. Amphipods thus lack specialized larval dispersal stages and the juvenile and adult food sources are the same.

The lack of external nourishment for embryos determines that the maximum biomass of oocytes in *F0* female approaching reproduction cannot be less than the biomasses of viable embryos. Maximum oocyte biomass in *FI* females is therefore less than the maximum oocyte biomass occurring in *F0* females bearing recently deposited embryos. In turn, maximum oocyte biomass in actively reproductive *FII* and *FIII* females cannot be less than maximum oocyte biomass in actively reproducing *FI* females. *F0* and *FIV* females lacking fully formed and maximum biomass oocytes thus cannot produce viable embryos and are

not ready for active reproduction. Males lacking complete spermatophores in addition to fully developed secondary sex characters are also not competent for mating.

Active reproduction in amphipods therefore requires synchronous gonad, oocyte, sperm and embryo development along with secondary sex morphologies. Starvation and trophic stress result in asynchronous development of these characters and delayed reproduction. The maturity of reproductive cells in amphipod gonads relative to brood maturity thus reveals the magnitude of energetic reserves and active reproduction or asynchronous and delayed reproduction. The same energetic constraints limit hatching juvenile biomass to the biomass of the embryos from which they formed. Moreover, due to their direct development, the smallest juveniles in summer are the progeny of the *FIV* females in their same populations. The minimum biomasses of juveniles in spring therefore reveal the maximum biomasses of winter oocytes and embryos from which they developed. Demchenko et al.'s (*2016*) default conclusions of summer starvation and winter growth and production are therefore testable, in part, from comparisons of the synchrony of *A. eschrichtii* reproductive morphologies and gonad development in summer and from comparisons of the reproductive characters in *A. eschrichtii* with the same characters in other amphipod species in the world during their periods of active reproduction.

## METHODS

Morphologies for pelagic mating, brood stages, and embryo development in amphipods are apparent directly (*Hastings, 1981*; *Sainte-Marie, 1991*; *Johnson, Stevens & Watling, 2001*; *Demchenko et al., 2016*). Gonad and reproductive cells are apparent from histology (*Charniaux-Cotton, 1985*; *Demchenko et al., 2016*). We tested for active summer reproduction in *A. eschrichtii* on the basis of six predicted characteristics that could be observed directly or resolved from histological preparations as follows:

1. all brood development stages present;
2. synchronous ovarian and brood development;
3. reproductively viable oocytes;
4. fully developed ovaries;
5. similar reproductive effort to other actively reproducing amphipods and;
6. mature sperm in males with fully developed secondary sex morphologies.

We interpreted evidence in support of these predicted characters and the coincidence of these characters with other amphipod species as evidence of active reproduction in summer. We interpreted the lack of evidence for these predicted characters or inconsistencies of these characters in *A. eschrichtii* with the same characters in other amphipod species as evidence of reproductive failure and starvation in summer. Failed predictions in our comparisons are thus, by default, evidence of failing trophic success in summer and thus active reproduction in winter.

We selected 40 reproductive size females and 14 reproductive size males (lengths 16 mm and greater) for our histological analyses. The females were collected in October 2013 and in July and October 2015 and the males were collected in October 2013. We did not find *FI* or *FIII* females among the major samples used to select particular specimens for these

analyses. Although insufficient for comparisons of populations on the scales summarized in *Demchenko et al. (2016)*, our expanded histological data permit detailed comparisons of oocyte development with *A. eschrichtii* length and brood development over time sufficient to additionally test for starvation and reproductive failure in summer.

We separated females and males from each collection date into six length groups, spanning approximately 3 mm each, for histology. The specimens were soaked in fresh water for 24 h, dehydrated, cleared in xylene and then infiltrated with melted paraffin. The paraffin was cooled into blocks that were cut into 10 μm thick sections for mounting onto glass microscope slides. Sections containing gonad tissue were stained using hematoxylin and eosin and permanently mounted under glass cover slips. The slides and additional dissected specimens for these analyses are deposited in the museum collections of the National Scientific Center of Marine Biology FEB RAS. We photographed the mounted sections to illustrate cell and tissue conditions and to permit measurements of reproductive cell and gonad dimensions using Videotest (http://www.videotest.ru; VideoTesT Ltd., St. Petersburg, Russia). A list of cell anatomy abbreviations in our figures are included in Table S1.

We assessed embryo diameters, brood development, secondary sex characters and body lengths using a stereomicroscope equipped with a calibrated micrometer and classified brood development by *Tzvetkova*'s (*1975*) stages *F0–FIV* as follows: *F0*—rudimentary oostegites lacking egg retention setae and no brood; *FI*—uncleaved embryos (eggs) in the marsupium enclosed by oostegites with fully developed embryo retention setae; *FII*—cleaved embryos; *FIII*—fully formed juveniles; *FIV*—developed oostegites with embryo retention setae and marsupium empty.

We measured body length from the anterior of the head to the base of the telson. We based our estimates of embryo and oocyte biomass and volumes on average diameters estimated from their average of lengths and widths (*Van Dolah & Bird, 1980*; *Nelson, 1980*; *Sainte-Marie, 1991*; *Johnson, Stevens & Watling, 2001*; *Charron et al., 2015*). We included observations of brood numbers and embryo dimensions from all available years to obtain the maximum possible sample size. We classified ovaries with normal, undamaged vitellogenic oocytes as undamaged, ovaries with mixtures of undamaged and lysed vitellogenic oocytes in the same ovary, as partial lysis and ovaries containing only lysed vitellogenic oocytes as total lysis.

We assessed oocyte viability from their diameters, development and structure and their estimated biomass relative to our estimated and observed viable embryo sizes. We tested for water uptake effects on our estimates of oocyte and embryo biomass by comparing observed embryo diameters with diameters estimated from biomasses of the smallest amphipods in our samples. We assumed for these estimates that the smallest amphipods are 0-age juvenile at the size occurring when they hatched. We checked the specific gravity value used in our biomass estimates by testing whether diameters of weight estimated embryos equaled our observed embryo diameters. Additionally, since 0-age summer juveniles are likely progeny of co-occurring *FIV* females, we used the *A. eschrichtii* 0-age juvenile weight in summer to estimate winter embryo and oocyte diameters.
**Table 1 Reproductive development stages.** Frequencies among females bearing vitellogenic oocytes by collection dates and length group.

| Lengths | October 2013 | | | July 2015 | | | October 2015 | | |
|---|---|---|---|---|---|---|---|---|---|
| | F0 | FII | FIV | F0 | FII | FIV | F0 | FII | FIV |
| 16–18 | 2 | 0 | 0 | 0 | 0 | 0 | 0 | 0 | 0 |
| 19–21 | 3 | 0 | 0 | 0 | 0 | 0 | 1 | 0 | 0 |
| 22–24 | 4 | 4 | 0 | 2 | 2 | 1 | 3 | 2 | 3 |
| 25–27 | 1 | 4 | 0 | 0 | 2 | 1 | 0 | 0 | 2 |
| 31–33 | 1 | 0 | 2 | 0 | 0 | 0 | 0 | 0 | 0 |
| Totals | 11 | 8 | 2 | 2 | 4 | 2 | 4 | 2 | 5 |

Our estimates of minimum viable embryo biomass require that the volume per weight (specific gravity) of an early stage, undifferentiated peracaridean crustacean embryo is approximately 1.146 g ml$^{-1}$ (*Spaargaren, 1979*) or, 1cc /1.146 g and thus, 0.8726 cc per g. Thus, we used length and weight relationships to estimated the *Ampelisca* oocyte diameter required to produce a viable embryo diameter ($D$) containing sufficient weight (g) to produce a minimum length ($L$) (0-age) *A. eschrichtii* juvenile. *Demchenko et al.*'s (*2016*) summarized *A. eschrichtii* weight per length where: g wt $= 1.49E-5*L^{3.0605}$. For simplicity, we estimate the volumes of the normally ellipsoidal oogonia, oocytes and embryos, by their average diameters. The weight of a zero-age juvenile thus converts to the volume ($V$) of a spherical oocyte by the relation:

$$V = 0.8726g = \frac{4}{3}\pi\left(\frac{D}{2}\right)^3. \tag{1}$$

By substitution, an oocyte or embryo diameter ($D$), required for a 0-age *A. eschrichtii*, is therefore:

$$D = 2\sqrt[3]{\frac{0.6545g}{\pi}}. \tag{2}$$

Our third estimate of A. *eschrichtii* embryo viability was relative to embryo biomasses (assessed from diameters) of other amphipod species of similar length ranges reported in the extensive summaries of (*Van Dolah & Bird, 1980*; *Nelson, 1980*; *Sainte-Marie, 1991*).

## RESULTS

We found only *F0*, *FII*, and *FIV* females (Table 1) as *Demchenko et al. (2016)* observed in their 2002–2013 samples. Frequencies of reproductive stages *F0*, *FII* and *FIV* were similar among collection dates and body lengths (Table 1). The lack of *FI* and *FIII* females, due to their possibly short durations, in all samples relative to *F0*, *FII* and *FIV* brood stages thus appears unlikely. The missing or vanishingly rare *FI* and *FIII* brood stages are consistent instead with reproductive stasis or failure, and counter to prediction 1.

The frequencies of ovaries with undamaged, partial and total lysis of vitellogenic oocytes were similar among collection dates (Table 2). Body lengths and reproductive development stages were also similar among collection dates (Table 3). Progressive reproductive development with time was therefore not apparent in our samples. Moreover,

**Table 2 Ovary conditions among years.** Frequencies of with undamaged, partially lysed, and total lysed oocytes and with atrophied or regenerated ovaries by date.

| Date | Undamaged | Partial | Total | Atrophied | Regenerated |
|---|---|---|---|---|---|
| October 2013 | 4 | 4 | 9 | 2 | 2 |
| July 2015 | 1 | 4 | 3 | 0 | 0 |
| October 2015 | 3 | 0 | 8 | 0 | 0 |
| Totals | 8 | 8 | 20 | 2 | 2 |

**Table 3 Ovary conditions among reproductive stages and length classes.** Frequencies of *A. eschrichtii* female length classes and reproductive stages containing ovaries with undamaged, partially lysed, and total lysed oocytes and with atrophied or regenerated ovaries brood stage.

| Stage | Size group, mm | Undamaged | Partial | Total | Atrophied | Regenerated |
|---|---|---|---|---|---|---|
| | 16–18 | 2 | 0 | 0 | 0 | 0 |
| | 19–21 | 0 | 0 | 4 | 0 | 0 |
| *F0* | 22–24 | 3 | 3 | 3 | 0 | 0 |
| | 25–27 | 0 | 1 | 0 | 0 | 0 |
| | 31–33 | 1 | 0 | 0 | 0 | 0 |
| | 16–18 | 0 | 0 | 0 | 0 | 0 |
| | 19–21 | 0 | 0 | 0 | 0 | 0 |
| *FII* | 22–24 | 2 | 2 | 3 | 1 | 0 |
| | 25–27 | 0 | 2 | 3 | 1 | 0 |
| | 31–33 | 0 | 0 | 0 | 0 | 0 |
| | 16–18 | 0 | 0 | 0 | 0 | 0 |
| | 19–21 | 0 | 0 | 0 | 0 | 0 |
| *FIV* | 22–24 | 0 | 0 | 4 | 0 | 0 |
| | 25–27 | 0 | 0 | 3 | 0 | 0 |
| | 31–33 | 0 | 0 | 0 | 0 | 2 |
| | Totals | 8 | 8 | 20 | 2 | 2 |

here and as follows, asynchronous and delayed development, degeneration and insufficient development in reproductive cells was apparent in a majority specimens examined.

Vitellogenic oocytes in 16–18 mm length female ovaries (Figs. 1A–1C) occurred within a single-layer of secondary follicular epithelium. These oocytes included undamaged (Figs. 1A, 1C) and lysed (Fig. 1B) cells. The lysed yolk accumulated in the lumen of the ovary next to the ovarian wall in contact with the amphipod circulatory system. The diameters of nuclei in follicular cells in contact with the lysed yolk increased from 0.010 mm to 0.016 mm (Fig. 1D). Only these cells were likely to have been "atretic", using the yolk. The nuclei diameters of follicular cells that were not in contact with the lysed yolk (located on the right side of the oocyte, sfc, Fig. 1D) did not change.

Two 23 mm *FII* females from October 2015 contained only undamaged oocytes while each of the 15 other *FII* females from all three collection periods contained damaged oocytes. July 2015 *FII* females contained undamaged vitellogenic oocytes (Fig. 2A) that were prevalent in the anterior ovary sections and a prevalence of disintegrating vitellogenic oocytes in posterior ovary sections (Fig. 2B). All vitellogenic oocytes of *FII*

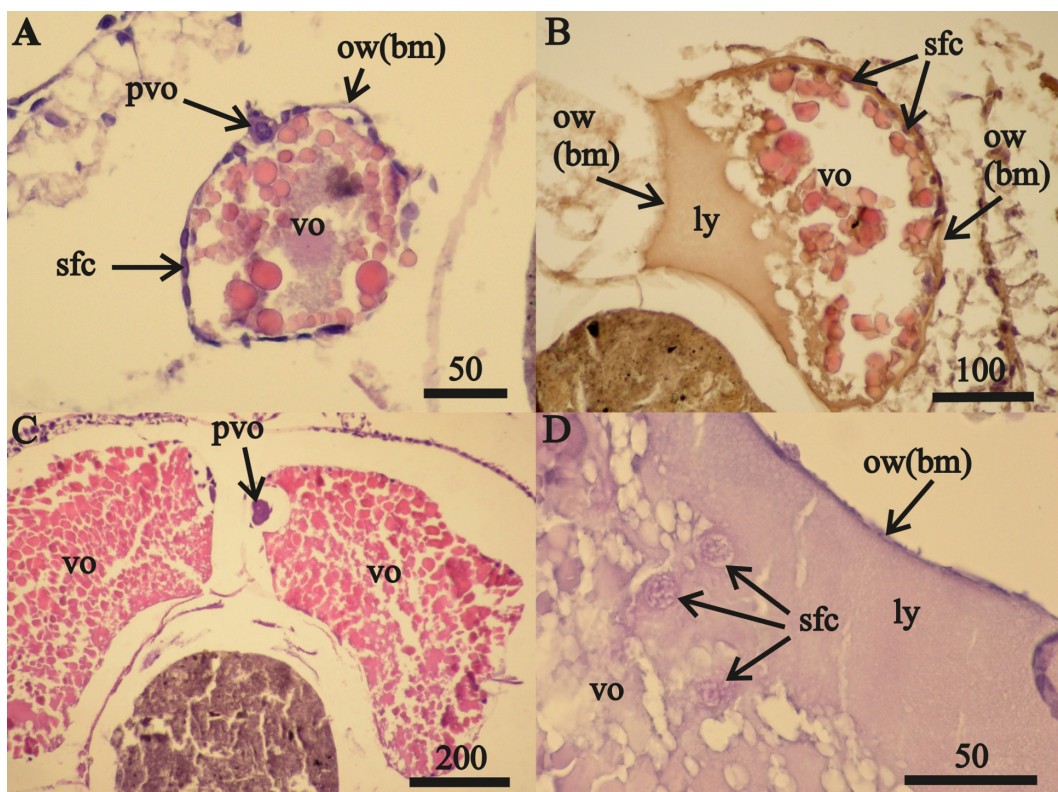

**Figure 1 Stage _F0 A. eschrichtii_ ovaries.** (A) Ovary of a 17 mm female with previtellogenic oocytes (pvo), vitellogenic oocytes (vo), secondary follicle cells (sfc); the ovarian wall (ow) is composed of the basal membrane (bm). (B) A 21 mm female undergoing lysis of vitellogenic oocytes revealing the lysed yolk (ly) that came out of the oocyte into the ovary lumen. (C) A 24 mm female with mature vitellogenic oocytes (vo). (D) Secondary follicular cells among lysed yolk adjacent to the ovarian wall (ow(bm)). All scales are in µm.

females from October 2013 and 2015 were undergoing lysis and resorption (Figs. 2C, 2D). Oocyte resorption was accompanied with mass mortalities of follicular epithelium cells. Hematoxylin did not stain the nuclei of these epithelial cells, which had swelled and then disintegrated (Fig. 2E).

Only expanded tubes of fibrous connective tissue (of basal membrane) and remnants of previtellogenic and vitellogenic oocytes remained in two (24 and 26 mm length) _FII_ females from October 2013 instead of functional ovaries (Fig. 2F). We classified the ovaries of these two females as atrophied (Tables 2 and 3).

The anterior ovary sections of one 32 mm _FIV_ female were reduced to empty tubes composed of the basal membrane (Fig. 3A). A germinal zone occurred in the middle ovary sections of this female that contained mesoderm cells and sparse, primary oogonia (Fig. 3B). The oogonia and nuclei diameters in this female were, respectively, 0.040 mm and 0.029 mm. Transformation of the mesoderm cells into follicular cells was apparent in their ovary germinal zones (Fig. 3C).

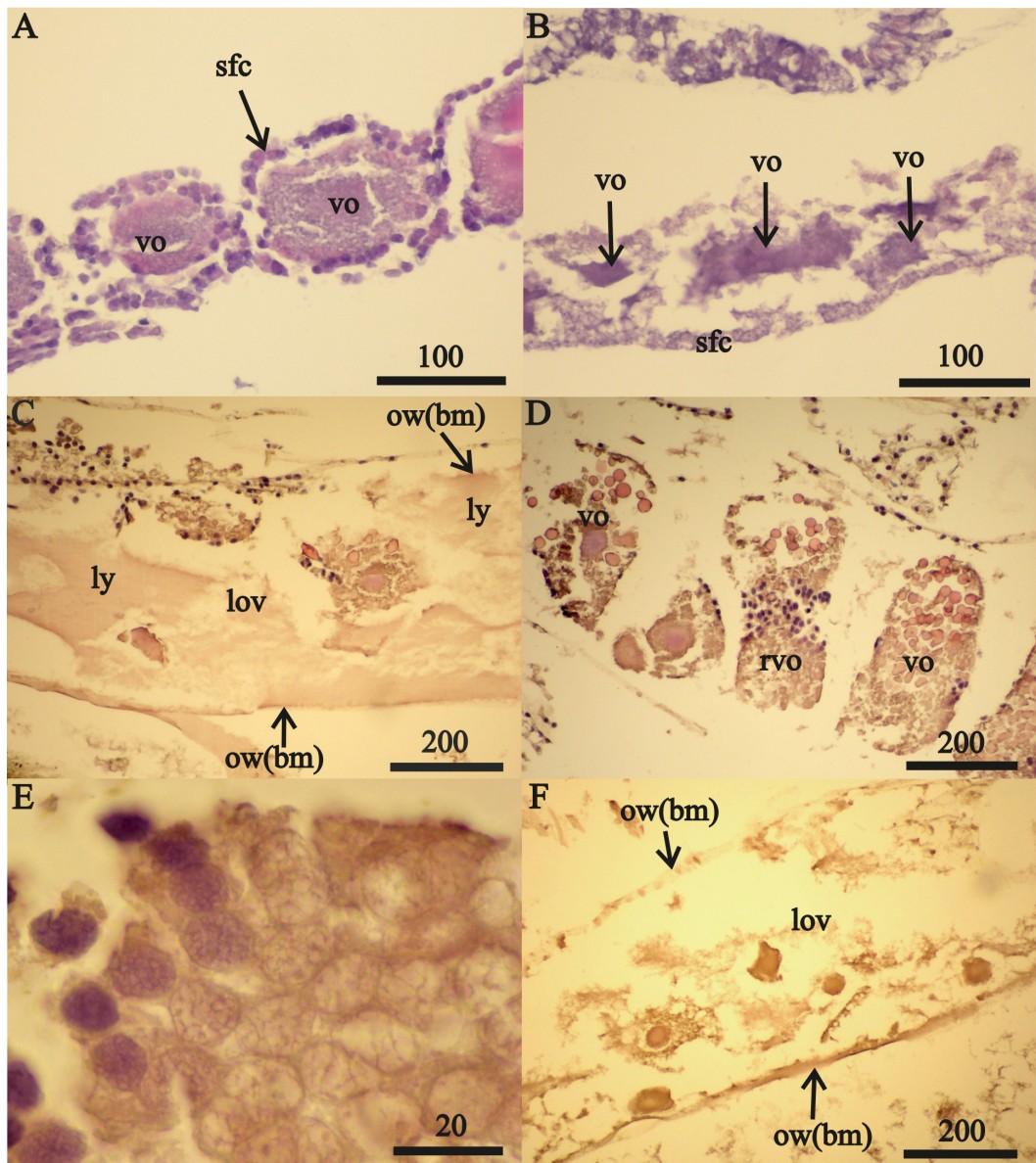

**Figure 2** **Stage *FII A. eschrichtii* ovaries.** (A) A 23.5 mm female with normal vitellogenic oocytes in the anterior section and (B) degraded vitellogenic oocytes in the posterior ovary section. (C) A 24 mm female with lysed yolk of oocytes inside of the ovary. (D) Resorption of vitellogenic oocytes by follicle cells (rvo). (E) Destruction of follicle cells in the process of resorption of vitellogenic oocyte. (F) Remnants of oocytes in the ovary lumen (lov). All scales are in μm.

The middle ovary sections of this female also contained oogonia in the prophase, anaphase and telophase of mitosis (Fig. 3D) and 0.026 mm diameter primary oogonia with 0.019 mm diameter nuclei (Fig. 3E). Posterior ovary sections included previtellogenic oocytes of variable sizes (Fig. 3F) that contained large granules of chromatin in their nuclei (first prophase of meiosis) and cells of primary follicular epithelium. The overall structure of this female's ovaries indicated they were recovering *de novo* after atrophy. Regeneration

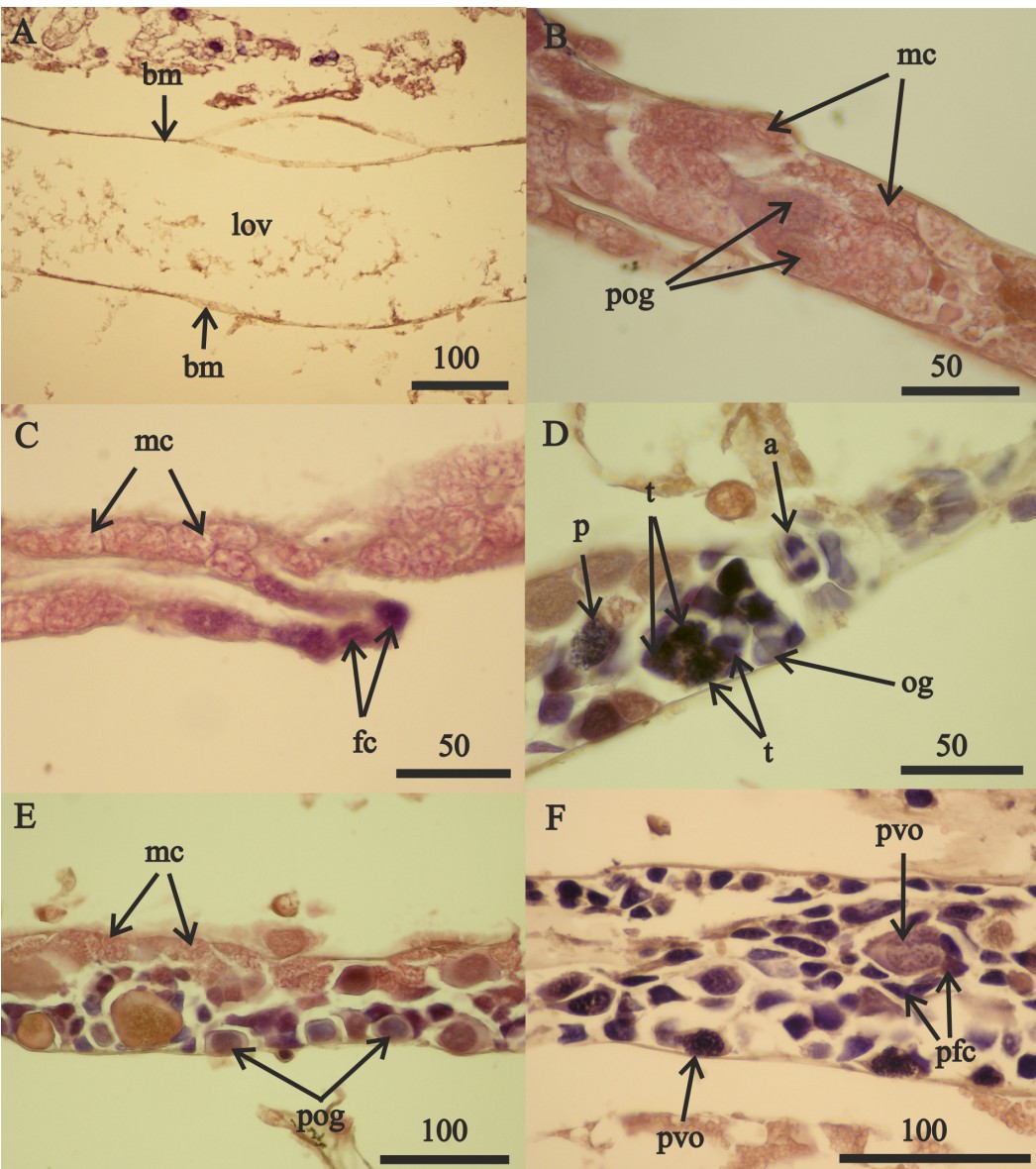

**Figure 3** **Stage *FIV* 32 mm *A. eschrichtii* ovary in *de novo* recovery.** (A) The empty anterior ovary section. (B) Large primary oogonia (pog) in the germinal zone. (C) Mesodermal cells (mc) transforming into follicular cells (fc). (D) Oogonia in anaphase, prophase and telophase of mitosis (a, p and t, respectively). (E) Primary oogonia in the germinal zone. (F) Posterior ovary section with previtellogenic oocytes. All scales are in µm.

of these ovaries was from the posterior end (opening into pereonite 5) and advancing to the anterior sections (near pereonite 2). The ovaries of the second 32 mm *FIV* female contained previtellogenic oocytes with large granules of chromatin in their nuclei in anterior sections and small vitellogenic oocytes that appeared to be new in posterior sections. The ovaries of these two 32 mm *FIV* females (Tables 2 and 3) therefore appear to have "regenerated".

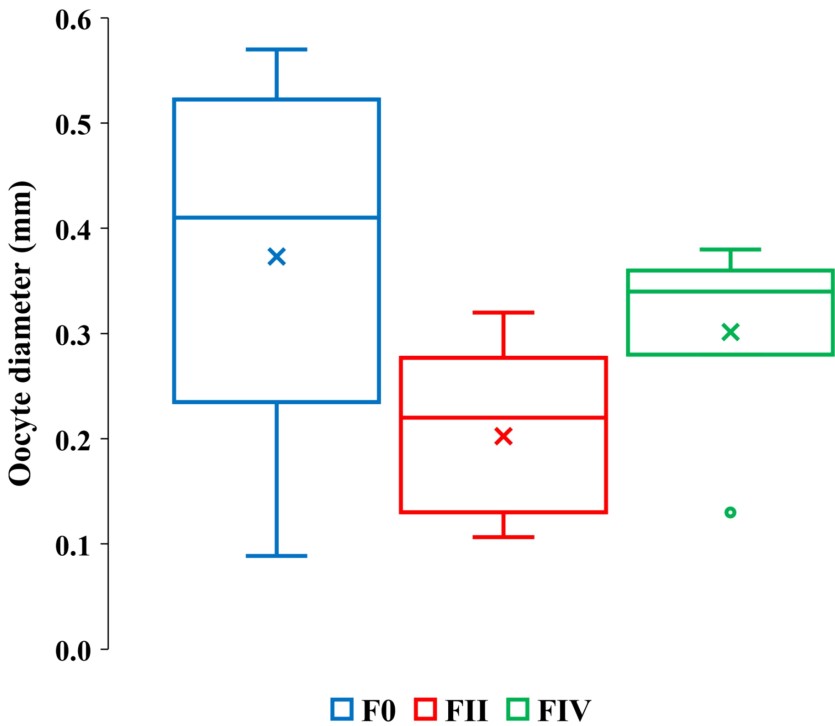

**Figure 4** **Quartile ranges of vitellogenic oocyte diameters with brood development.** Range, upper and lower quartile (box) mean (x), and median (solid line) of vitellogenic oocyte diameters in *F0*, *FII* and *FIV* females (*N* = 17, 13 and 7, respectively).

A 32 mm *F0* female (Table 3) contained 0.490 mm diameter vitellogenic oocytes. The undeveloped oostegites of this female indicate that she was also recovering from a non-reproductive period but her large oocytes place her at an advanced undamaged reproductive condition in contrast to the two other 32 mm (stage *FIV*) females.

Eight *F0* and *FII* females ranging in lengths between 16 to 32 mm contained only undamaged vitellogenic oocytes (Table 3, column 3). The length ranges of these females were overlapped by 28 *F0*, *FII* and *FIV* females ranging from 19 to 27 mm in length with partially or totally lysed vitellogenic oocytes (Table 3, columns 4–5). Our sample size was insufficient to resolve whether the frequencies of total or partially lysed vitellogenic oocytes varied significantly between *F0* and *FII* and *FIV* females or how their brood stage frequencies vary relative to other reproductive amphipods. However, a greater range of vitellogenic oocyte diameters occurred among the 16 to 32 mm *F0* females than among the 22 to 32 mm length *FII* and *FIV* females (Fig. 4, Table S2).

The greater diameter oocytes among *F0* and *FIV* females than among *FII* females (Fig. 4) are consistent with expected increases in oocyte growth with reproduction. *FIV* females in actively reproductive populations, however, are ready for a new brood and therefore expected to bear the largest sizes of oocytes. The oocytes of our *FIV* females were thus not large enough to produce viable embryos and out of synchrony with their brood development, counter to prediction 2.
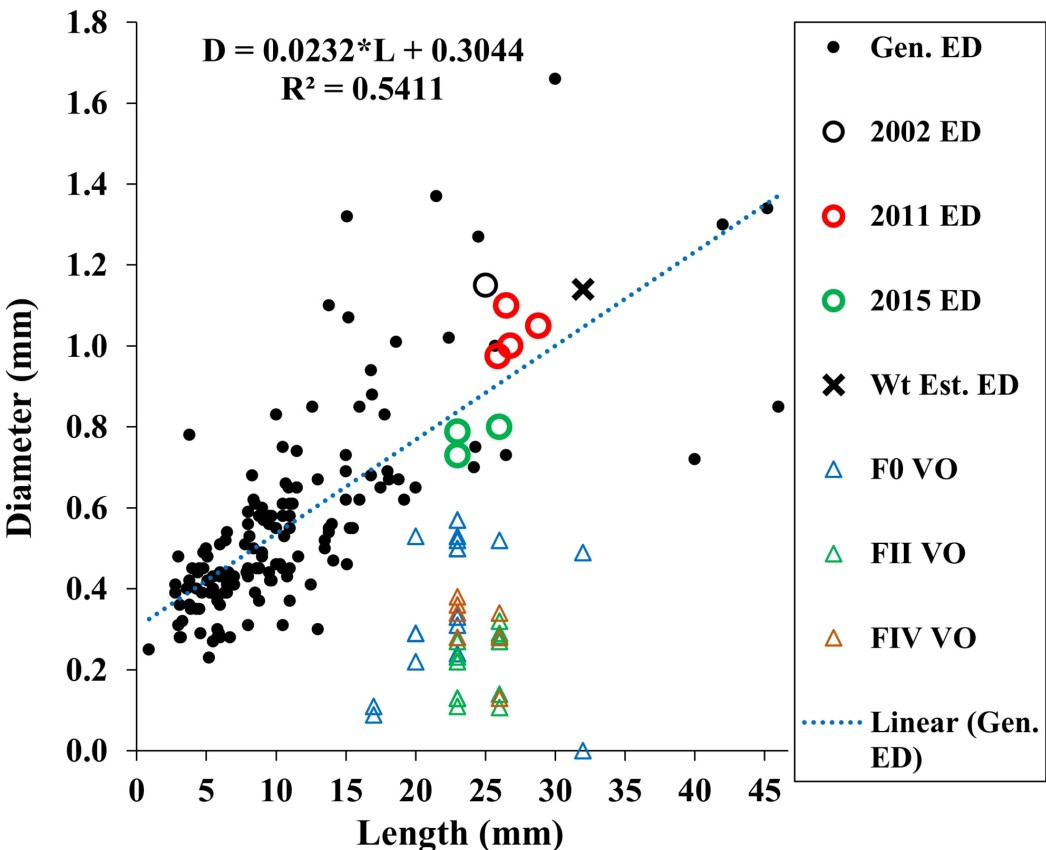

**Figure 5  Maximum vitellogenic oocyte and embryo diameters with body length among amphipod populations.** Embryo diameters (ED) with body length among 123 gammaridean amphipod species of the northern hemisphere (black dots, *Sainte-Marie, 1991*, appendix Table 1), Sakhalin Shelf *A. eschrichtii* embryo diameters from 2002 (black circle), 2011(red circles) and 2015 (green circles), estimated embryo diameter for a 3.8 mm juvenile *A. eschrichtii* (black X) and observed oocyte diameters for *F0*, *FII* and *FIV* females (blue, green and red triangles, respectively).

Our estimated minimum embryo diameter (from the weight of the smallest, 3.8 mm length, *A. eschrichtii* that we found in our samples) was 1.14 mm. The 2002, 2011 and 2015 *A. eschrichtii* embryo diameters (Fig. 5, black, red and green circles), ranged between 0.76 and 1.15 mm and were from females averaging 23.6 mm in length. Our observed *A. eschrichtii* embryo diameters (Fig. 5) are also within the range of embryo diameters expected for a 23.6 mm generalized amphipod (Fig. 5, equation). Our estimated embryo diameter is thus within the range of diameters observed in *A. eschrichtii* herein and other similar sized gammaridean amphipod species (*Sainte-Marie, 1991*, Fig. 5).

Oocyte diameter sufficient to produce a viable juvenile *A. eschrichtii* (prediction 3) must equal or exceed embryo diameters. However, all oocyte diameters in our samples were less than the observed or estimated minimum diameter embryos (Fig. 5). Thus, counter to prediction 3, our specimens did not have oocytes suitable to produce viable sized embryos.

Counter to prediction 4, two 32 mm *FIV* females had apparently "regenerated" ovaries and the ovaries of two *FII* females (24 mm and 26 mm in length) were atrophied (Tables 2

and 3, Fig. 2F). The anterior ovary sections of one of *32* mm female were reduced to empty tubes composed of the basal membrane (Fig. 3A). Compromised ovaries thus occurred in a wide size range and two brood stages of reproductive females. Persistence of females that cannot later reproduce is likely to be maladaptive and thus an evolutionary conundrum. The atrophy of ovaries during poor trophic conditions and the resorption of reproductive cells is thus more likely to be adaptive for increasing survival until trophic conditions improve.

We used the antilog of *Sainte-Marie*'s (*1991*, Table 9) estimated ampeliscid brood size (BS) with body length (BL): [$BS = 1.227 \star BL^{1.335}$, $r^2 = 0.49$, $n = 24$)] to compare with *A. eschrichtii* (Fig. 6). The correlation of amphipod embryo size with body length (*Sainte-Marie, 1991*) and similar winter and summer embryo sizes in *A. eschrichtii* reveal a nearly constant relation between embryo and amphipod size. The constant embryo to body size ratio permits direct comparisons of reproductive effort from the number of embryos per amphipod length. Our samples, consisting of one brood from 2002, four broods from 2011 and fifteen broods from 2015 (Fig. 6), were respectively, 34%, 15% and 49% of the expected size adjusted ampeliscid brood size. (Note that the observed embryo with body length equation (2015 BS) (Fig. 6) includes only the July and September 2015 populations.) The brood sizes of *A. eschrichtii* in summer are less than the expected among amphipods in general and are thus evidence of reduced reproductive effort in summer, counter to prediction 5.

Reproductive development advanced in males with length (Fig. 7) and the spermatophores present in greater than 21 mm in length, male testes indicate reproductive competence. The testes primordia (two narrow cords of mesoderm cells [mc]) occurring in 16.5 mm length males (Fig. 7A) and rare spermatogonia with nuclei that stained with hematoxylin, occurred only on the periphery of the cords. The testes of 18 mm males, in addition to the mesoderm cells, contained well developed germinal zones with spermatogonia and spermatocytes outside the germinal zone (Fig. 7B). Testes of 20 mm males also contained spermatocytes and spermatids (the product of meiotic division of spermatocytes) in the lumen. Testes of 21 mm males also included accessory cells (Fig. 7C) that are associated with the transformation of spermatids into spermatozoa (*Charniaux-Cotton, 1985*) and the seminal vesicles of these males contained numerous spermatozoa (Fig. 7D). The *vas deferens* of these 21 mm males contained spermatophores that were packed with spermatozoa (Fig. 7E).

The testes of greater than 21 mm males lacked germinal zones and the testes walls of these males were sparsely lined with mesodermal cells and thus were no longer capable of producing additional sperm. Testes of 24 and 26 mm males contained few spermatocytes or spermatids. The flattened accessory cells and rare mesodermal cells of the testes of these males (Fig. 7F) indicate they were not producing additional sperm cells. The spermatozoa in the seminal vesicles and spermatophores in the *vas deferens* of the greater than 21 mm males were fertile and possibly capable of mating. However, fully developed secondary sex morphologies required for pelagic mating (*Hastings, 1981*) were lacking among all sizes of males examined. Thus, males incompletely satisfied prediction 6 (mature sperm and developed secondary sex morphologies).

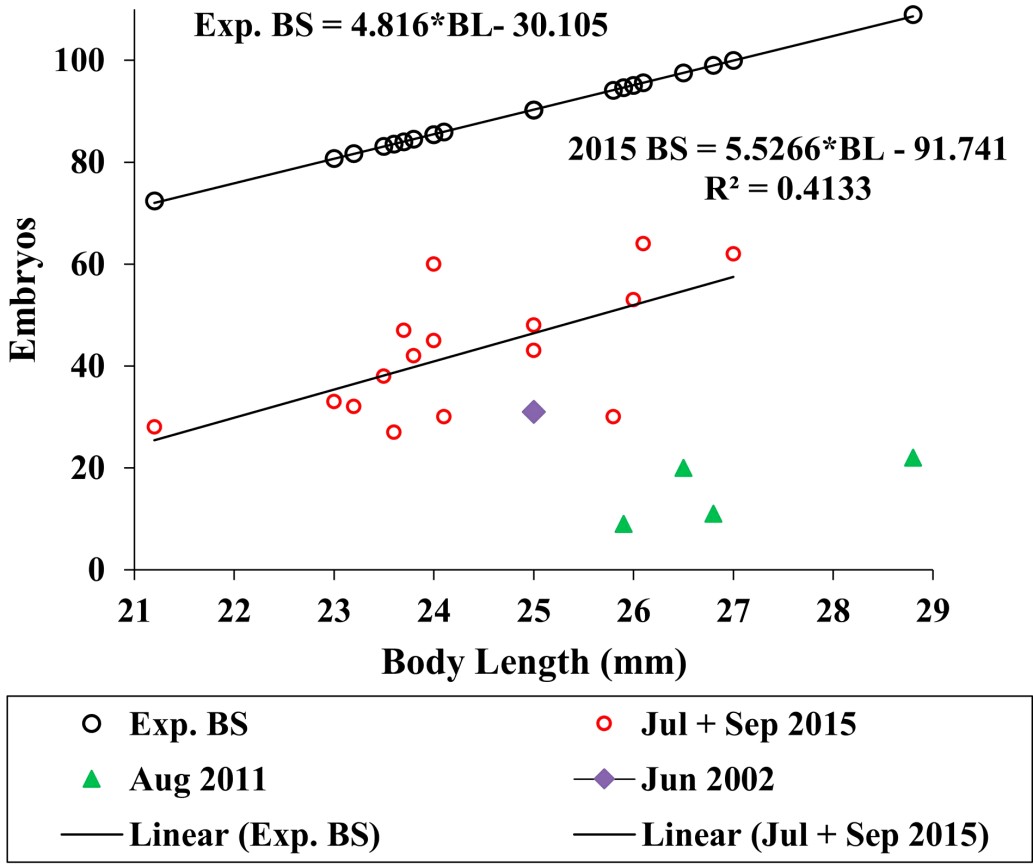

**Figure 6** **Expected and observed brood sizes.** Expected brood sizes (Exp. BS) (black line and circles) and observed *A. eschrichtii* brood sizes of July and October 2015 (2015 BS) (red circles, $N = 16$), June 2002 (purple diamond, $N = 1$) and August 2011 (green triangles, $N = 4$) with body length (BL).

## DISCUSSION

Our observations are counter to six predicted characteristics of active summer reproduction. None of the 40 females and 14 males were ready for active reproduction. The absence of *FIII* females is of particular interest when *FIV* females were present. The proportions of *FIV* females relative to other brood stages must decline as the time between juvenile release and molting (that immediately precedes deposition of new embryos) decreases. The combined frequency of *FI* and *FIII* females is therefore expected to greatly exceed the frequency of *FIV* females during active reproduction. However, consistent with *Demchenko et al.*'s (*2016*) observations (based on a larger sample size), and counter to prediction 1, we did not find any brood stages *FI* and *FIII*.

Maximum size oocytes are expected in all females that are ready to deposit new embryos. We therefore expected to find maximum diameter oocytes in some *F0* females and all *FIV* females if they were actively reproductive. We also expected steadily increasing maximum oocyte diameters from *FI* to *FIV* females. The more recently deposited embryos of stage *FII* females depleted the largest oocytes from the ovaries of these females. The reduced

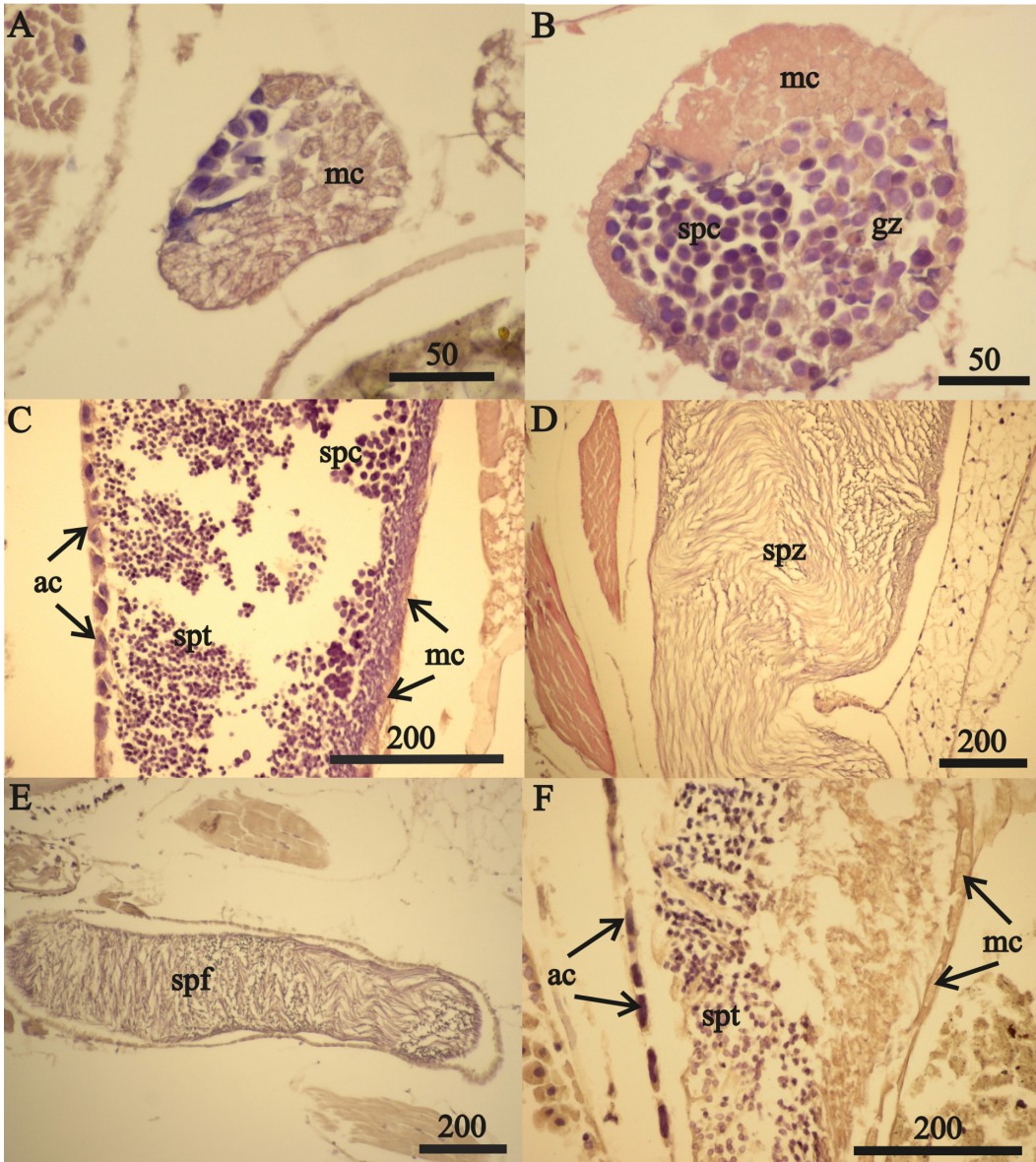

**Figure 7** *A. eschrichtii* **testes.** (A) Cord of mesodermal cells (mc) of a 16.5 mm male previous to functional testis. (B) Germinal zone (gz) and spermatocytes (spc) of a 18 mm male testis. (C) Accessory cells (ac), spermatocytes (spc) and spermatids (spt) of a 21 mm male testis. (D) Spermatozoa (spz) in seminal vesicle of a 21 mm male. (E) Spermatophore (spf) in *vas deferens* of a 21 mm male. (F) Atrophied spermatids (spt) of a 26 mm male testis. All scales are in μm.

maximum size oocytes in *FII* females is therefore consistent with prediction 2. However, counter to prediction 2, the maximum oocyte diameters in *FIV* females were not as great as in *F0* females. The oocytes of *FIV* females were too small to produce viable embryos. The small oocytes of *FIV* females therefore indicate reproductive asynchrony, reproductive stasis and food stress. They are counter to prediction 2.

The high prevalence of lysed oocytes among all brood stages in 28 of the 40 females and in all sample periods are counter to prediction 3. Atrophied ovaries of the 24 and 26 mm *FII* females and the two 32 mm *FIV* females indicate use of the content of the ovaries, through lysis and resorption, for energetic needs. These females were sufficiently large to have produced previous broods and co-occurred with similar size females with disintegrating oocytes. Reduced ovaries of these females are inconsistent with synchronous reproduction and with prediction 4. The atrophy of ovaries may be an extreme adaptation of *A. eschrichtii* to starvation. The presence of a 32 mm *F0* female with large vitellogenic oocytes without signs of lysis indicates the successful functioning of the restored ovaries that is likely to be a relatively lengthy process.

Testing prediction 5 depends on whether the relatively low observed brood numbers of *A. eschrichtii* is a result of reduced reproductive effort (brood biomass per weight of female). Water uptake with the conversion of yolk reserves into structural elements did not appear to increase the *FII* embryo dimensions or otherwise confound our results. These embryos had not expanded significantly (*Sheader, 1996*). The lower than expected *A. eschrichtii* embryo numbers were thus not compensated for by larger embryo sizes. These data indicate that juvenile *A. eschrichtii* biomass is similar to other similar sized amphipod species and that the biomasses of individual *A. eschrichtii* embryos in summer are sufficient to produce the smallest observed juveniles from winter. Thus, size differences between winter and summer embryos, that could confound estimates of reproductive effort, appear unlikely. The low embryo counts thus represent low reproductive effort of *A. eschrichtii* in summer relative to similar sized amphipod species and are therefore counter to prediction 5.

The mature sperm in the *vas deferens* of the largest males are consistent with active summer reproduction (prediction 6). Sperm are not rich in lipids and thus, are minor energy sources or thus indicators of trophic stress. We assume that reproductive investments of males are greater into somatic tissues than into gonads and reproductive cells. However, counter to prediction 6, males with terminal phase pelagic mating morphologies did not occur in these samples or any previous summer samples (*Demchenko et al., 2016*).

## CONCLUSIONS

*Ampelisca eschrichtii* appear to absorb their oocytes, delay reproduction and their reproductive effort is less than expected in summer. None of the 40 females examined were competent to reproduce. The low embryo counts are consistent with cannibalism under starvation conditions, as observed in other amphipods (*Sheader, 1996*; *Hyne, 2011*). *Sheader (1996)* experimentally demonstrated embryo losses due to cannibalism in *Gammarus insensibilis* and that the oocytes of females that do not ovulate are resorbed. Oocyte lysis and resorption are thus likely to be common responses of amphipods to starvation. The *FII* and *FIV* females in our samples had therefore possibly devoured some and all of their embryos, respectively. However, embryo cannibalism would be the most severe of all responses to starvation. We did not compare ovaries and brood counts of the same females but assume cannibalism does not begin until all vitellogenic oocytes and possibly the ovaries also are absorbed.

These results corroborate *Demchenko et al.*'s (*2016*) previous conclusions of starvation, reduced growth and reproduction delays in summer. There is no evidence of hypoxia (*Rutenko & Sosnin, 2014*) or massive redistribution of sediments in summer on the Sakhalin Shelf that would appear likely to restrict growth of these mobile suspension feeding amphipod populations. *Coyle et al.*'s (*2007*) proposal, that growth and production of *Ampelisca macrocephala* Liljejborg, 1852 in the Bering Sea occurs in summer, is counter to our conclusion of winter growth and production by *A. eschrichtii* on the Sakhalin Shelf. However, winter production and adaptations to low temperatures and to winter growth and reproduction are prevalent among high latitude amphipods (*Bregazzi, 1972*; *Kusano, Kusano & Watanabe, 1987*; *Jakob et al., 2016*). Amphipod breeding seasons, revealed by maximum brood sizes (*Morino, 1978*; *Sheader, 1983*; *Charniaux-Cotton, 1985*; *Geffard et al., 2010*), by oogenesis and embryo production (*Charron et al., 2015*) and by juvenile releases (*Sagar, 1980*; *Sutcliffe, 1993*) coincide with maximum food resources. *A. macrocephala*, the most closely related species to *A. eschrichtii*, are similarly distributed around the northern hemisphere (*Dauvin, 1988*; *Barnard & Karaman, 1991*), release their juveniles in coincidence with maximum phytoplankton abundances and can survive for at least 5 months in aquaria without food (*Kanneworff, 1965*). High latitude *Ampelisca* thus appear to have adaptations for starvation and to reproduce in coincidence with food abundance rather than with season or temperature. Reduced growth and reproduction of *A. eschrichtii*, spanning multiple years and months of summer is a likely adaptation to trophic stress and starvation. Thus, our default conclusions from these data are that *A. eschrichtii* starve in summer and feast in winter.

Trophic stress among Sakhalin Shelf *A. eschrichtii* populations in summer is also consistent with Sakhalin Shelf oceanography. Phytoplankton biomass, including diatoms, is concentrated in summer over the Sakhalin Shelf at the upper boundary of a thermocline ranging from the surface to 10–15 m depths (*Sorokin & Sorokin, 1999*; *Rutenko & Sosnin, 2014*; *Prants et al., 2017*). Vertical mixing and down-welling of Sakhalin Shelf waters is prevalent in winter (*Leonov et al., 2007*). The 40-60 m depth ranges of the Offshore benthos are thus below the high surface concentrations of phytoplankton in summer. These benthic populations, that occur below 10 m, are more likely to receive phytoplankton when winter storms mix the water column and abundant surface phytoplankton to their depths. Our default conclusions remain open to direct tests that should include surveys of these amphipod populations in the Offshore gray whale feeding area in winter. These direct tests would resolve the life history and ecology of this critical western gray whale prey source and would also provide a major contribution to the global understanding of high latitude benthic community ecology and production.

## ACKNOWLEDGEMENTS

VBD, JWC and NLD accept all responsibility for the integrity and validity of the data collected and data analyses. We thank EP Dats (Vladivostok State University of Economics and Service, VSUES) for assistance with software, formula editing and calculations. We thank Dr. Gilbert Rowe (Department of Oceanography, Texas A&M University) and an

anonymous reviewer and for their helpful reviews. We thank Dr. Ralph A. Breitenstein, Hatfield Marine Science Center, CEOAS, Oregon State University, for proof reading a late version of this manuscript. We thank Dr. Vladimir Efremov (WGW/MMPP Coordinator, ENL) for encouraging us in the production of the manuscript.

### Funding

Exxon Neftegas Limited and Sakhalin Energy Investment Company funding under the joint gray whale monitoring program supported all field work under which all Sakhalin Shelf samples reported herein were collected and also provided the publication funds for this report. Exxon Neftegas Limited and Sakhalin Energy Investment Company had no role in study design, data analysis, decision to publish, or preparation of the manuscript.

### Grant Disclosures

The following grant information was disclosed by the authors:
Exxon Neftegas Limited.
Sakhalin Energy Investment Company.

### Competing Interests

The authors declare there are no competing interests.

### Author Contributions

- Valentina B. Durkina, John W. Chapman and Natalia L. Demchenko conceived and designed the experiments, performed the experiments, analyzed the data, contributed reagents/materials/analysis tools, prepared figures and/or tables, authored or reviewed drafts of the paper, approved the final draft.

### Data Availability

Additional information and raw data for Tables 1–3 and Figs. 5 and 7 are included with the Supplemental Information.

### Supplemental Information

Supplemental information for this article can be found online at http://dx.doi.org/10.7717/peerj.4841#supplemental-information.

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
