# Peer review of "Ampelisca eschrichtii Krøyer, 1842 (Ampeliscidae) of the Sakhalin Shelf in the Okhotsk Sea starve in summer and feast in winter"

_PeerJ, doi:10.7717/peerj.4841_

## Round 0.1 · original submission · Major Revisions

· Academic Editor

Major Revisions

The referees really liked your topic, as I did. However, they feel the paper needs a lot of work with one recommending rejection and the other revision. As both agree there is data here worth publishing I recommend Major Revision which will mean more work for the referees. If you decide to revise please make your best effort to improve the paper to makes the referees next review easy. If you decide not to resubmit please withdraw the paper.

·

Basic reporting

No comment

Experimental design

No commebt

Validity of the findings

No comment

Additional comments

Review of Demchenko et al. ms. on A. eschrichtii ms. for PEERJ, ms. 22763.

This is a fine ms. First let me address its importance: big amphipods that are fed upon by whales is a rate example of benthic - pelagic coupling in the form of predation. The more we know about these big amphipods, the better. Most work in BPC is concerned with recycling of nutrients back to the water column, but this ecosystem is more tightly coupled than most.This is direct transfer of carbon, nitrogen and energy.

This ms. is trying to demonstrate, indirectly, that the amphipods grow and reproduce in the winter rather than the summer. The evidence is good that little is happening in the summer, based on their delightful histology. The big question remaining is why they did not sample in the winter and spring? They ought to tell the reader why this obvious approach was not done. Ice? Too rough? No ship?

I like their comment in the short discussion section that the diatoms get to the bottom when the water mixed in the winter but not in the summer when the water is stratified. But could other things be going on too? How about hypoxia in the summer due to the stratification? Erosion?

Bottom line: this paper could be published more or less as is. My comments above are meant to be helpful, not a detraction.

Here are some things I noted: Fig 5C, is the 'spc' on the figure actually named 'spt' in the caption?
line 230, drop the comma
line 231, 'these greatest' should be 'of these'
line 237, change 'delayted' to 'delayed'
lin e 248, drop the comma
line 283, drop the comma
line 383, in the references, all caps in title should be lower case.

Reviewer 2 ·

Basic reporting

The manuscript as it stands lacks from structure with elements of the methods and results more suited to the introduction and discussion. For example, the methods starts with a series of predictions/hypotheses with their underlying rational which arguably should have been laid out in the introduction. The methods then leaps into a list of life history stages without the context for the list stated when the methods/dates/location relating to the collection of the animals would be been more appropriate.

The results section is quite difficult to follow in places and is a description of histology would be helped if put in the context of the study and have a narrative to aid the reader through. Some of the results section could be rewritten in an easier to read format (e.g. page 6 line 187) "We classified their ovaries as atrophied" doesn't refer to anything and the next sentence has intro material and not results.

Experimental design

The experimental design is quite hard to assess as the number of animals collected and used for each element of the histology is hard to determine. On that basis it is hard to determine that the conclusions reached are valid given the value of 'n' and the lack of statistical analsyses. This makes the conclusions and the title rather speculative at this stage.

Validity of the findings

As mentioned above and highlighted by the authors (e.g. page 13 line 287) "Atrophied ovaries of two FII females indicate starvation and maximum use of the content of their ovaries..........". It appears to this reviewer that their is quite a lot of speculation in this paper based on a very low sample size from which the authors have based their entire conclusions and set their title of the paper.

---

## Round 0.2 · accepted · Accept

· Academic Editor

Accept

Thank you for revising the MS. One of the referees was able to study and is pleased with the changes. Thank you for considering PeerJ.

# ·

Basic reporting

None

Experimental design

none

Validity of the findings

None

Additional comments

Review of (rather re-review of) ms. for PeerJ on Ampeliscid gonad status and what it means (ms. 22763v2)

I stand by my original conclusion: the ms. is now improved with more information. This is an important subject. It is very difficult to determine seasonality in organisms in the field. They have some evidence of it from histology. That is commendable.

I do not understand or agree with the second reviewer's comments. He or she makes no mention of the importance of the observations, only that they are too few from which to make conclusions. The earliest observations of seasonality in the deep benthos were scant but nonetheless important and now have proved to be quite valid. So, as I said above, publish this paper.